# Advances and Challenges in the Treatment of HPV-Associated Lower Genital Tract Cancers by Immune Checkpoint Blockers: Insights from Basic and Clinical Science

**DOI:** 10.3390/cancers17081260

**Published:** 2025-04-08

**Authors:** Marhama Zafar, Narjes Sweis, Hitesh Kapoor, Gerald Gantt

**Affiliations:** Division of Colon and Rectal Surgery, University of Illinois at Chicago, 840 S Wood Street, Chicago, IL 60612, USA; nsweis5@uic.edu (N.S.); hk96@uic.edu (H.K.); ggantt2@uic.edu (G.G.)

**Keywords:** HPV, anal cancer, cervical cancer, vulvar cancer, penile cancer, vaginal cancer, immunotherapy

## Abstract

The rising number of HPV-related lower genital cancers emphasizes the need for better awareness, prevention, and treatment strategies. The use of immunotherapy in treating HPV-related lower genital cancers provides a hopeful avenue for improving patient outcomes. This review takes a close look at the challenges of HPV-related cancers and highlights the recent progress in their treatment. We also cover ongoing research and clinical trials that are important for improving immunotherapy for patients with HPV-related cancers.

## 1. Introduction

Human papillomavirus (HPV)-related lower genital cancers include cervical cancer, anal squamous cell carcinoma (SCC), vaginal cancer, vulvar cancer, and penile cancer. Approximately 45,000 new cases of HPV-related tumors are diagnosed annually, with significant morbidity and mortality [1]. These cancers are linked to persistent infection with oncogenic HPV types 16 and 18.

Current treatment options for these cancers include a combination of chemoradiotherapy, systemic chemotherapy, and immunotherapy. Immune checkpoint inhibitors (ICIs) are a class of immunotherapeutic drugs that block proteins used by cancer cells to escape immune surveillance, thereby enhancing the body’s immune response to cancer cells. ICIs have emerged as promising options for advanced or metastatic cancer in several tumor sites.

This narrative review summarizes the changes in the tumor microenvironment of HPV-associated cancers, the mechanisms of action of immune checkpoint inhibitors, their efficacy in treating these tumors, and the challenges involved. It synthesizes key preclinical and clinical advances in the use of immune checkpoint inhibitors for HPV-associated lower genital tract cancers. To supplement this discussion, we compiled a table of representative clinical trials selected through a targeted literature search using PubMed and ClinicalTrials.gov. Keywords included but were not limited to “HPV”, “immune checkpoint inhibitors”, “cervical cancer”, “vaginal cancer”, and “vulvar cancer”. Trials were selected based on their relevance to HPV-associated cancers and the use of immune checkpoint inhibitors (Table 1).

## 2. HPV-Associated Cancers

Cervical cancer is the most recognized HPV-associated cancer, with nearly all cases caused by HPV infection. The overall incidence of HPV-related cancers for women was 13.68 per 100,000, with cervical cancer accounting for more than half (52%) of these cases, at an incidence rate of 7.12 per 100,000 in 2017 [40]. HPV16 and HPV18 are responsible for approximately 70% of cervical cancers worldwide [41]. Persistent infection with HPV upregulates the expression of oncogenes E6 and E7, which inhibit the tumor suppressors p53 and Rb, respectively [42]. This causes the progression of initial infection to precancerous lesions, ultimately leading to the development of invasive cancer if not treated.

HPV is also associated with 90% of anal cancers, along with a significant proportion of vaginal, vulva, and penile cancers [43]. The rates of anal SCC increased among both men (2.1%) and women (2.9%) from 1999 to 2015 [1]. Over the past 16 years, there has been a consistent annual increase in anal and rectal SCC, with an annual percent change (APC) of 1.71% [40]. For vaginal, vulvar, and penile cancers, approximately 30% of tumors diagnosed in the United States are due to carcinogen exposure or other risk factors [44].

The impact of HPV vaccination is substantial as the incidence of cervical cancer in the United States has declined since the introduction of effective screening and vaccination programs. From 2001 to 2017, the incidence decreased, with an annual percent change (APC) of 1.03% [40]. A meta-analysis revealed a 51% reduction in the incidence of cervical intraepithelial neoplasia grade 2 + (CIN 2+) among girls aged 15–19 years and a 31% decrease among women aged 20–24 years [45]. While the HPV vaccine is a crucial preventative measure and has helped reduce the incidence of HPV-associated cancers, untreated cases lead to HPV, creating an immune-suppressive microenvironment. However, the understanding of the mechanism by which HPV alters the microenvironment is still limited.

## 3. Tumor Microenvironment and Mechanisms of Immune Checkpoint Inhibitors

The use of ICIs in treating HPV-associated cancers is grounded in their mechanism of interaction within the tumor microenvironment.

The tumor microenvironment is characterized by an accumulation of myeloid-derived suppressor cells (MDSCs) and tumor-associated macrophages (TAMs). These cells inhibit the activation of CD8+ T cells to decrease the anti-tumor immune response. One of the most extensively studied immune checkpoints is the interaction between programmed cell death protein 1 (PD-1) and its ligand, programmed death-ligand 1 (PD-L1). High-risk HPV oncoproteins, like E6 and E7, facilitate immune evasion by upregulating the interaction between PD-1 on T cells and PD-L1 on tumor cells [46]. Increased expression of PD-L1 has been observed in HPV-positive cervical cancer [47]. In anal SCC, PD-L1 expression has been observed in both HPV-positive and HPV-negative cases [48]. In another study, strong PD-L1 expression was more prevalent in tumors with a high HPV viral load, suggesting a correlation between HPV infection and PD-L1 expression [49]. Additionally, both tumor cells and tumor-associated macrophages in HPV-related head and neck cancers have shown overexpression of PD-L1, creating immune resistance [50]. Therefore, ICIs that disrupt the PD-1/PD-L1 interaction can improve therapeutic outcomes by boosting the immune response.

In addition to PD-1/PD-L1, the immune checkpoint protein cytotoxic T-lymphocyte antigen 4 (CTLA-4) acts during the priming phase of T cell development. HPV induces the expression of CTLA-4 on epithelial cells through its E7 oncoprotein, and in the context of HPV-related head and neck cancers, CTLA-4 expression is significantly elevated on TILs, which contributes to an immunosuppressive TME [51].

Indoleamine 2,3-dioxygenase 1 (IDO1) is an enzyme that plays a critical role in immune suppression within the tumor microenvironment of HPV-associated cancers. It breaks down tryptophan into kynurenine, resulting in local tryptophan depletion. This depletion causes the accumulation of immunosuppressive metabolites, which ultimately inhibits T cell function and enhances the activity of regulatory T cells (Tregs) [52]. In HPV-associated cancers, such as cervical cancer, IDO1 expression is upregulated and correlates with HPV antigen expression. This upregulation contributes to T cell exhaustion and dysfunction. IDO1 upregulation also increases the expression of PD-1, further impairing CD8+ T cell activity. In preclinical models, IDO1 inhibition has been shown to reduce the tumor volume, suggesting a therapeutic strategy to enhance the efficacy of immunotherapies in HPV-associated cancers [53].

Lymphocyte-activation gene 3 (LAG3), another immune checkpoint found on activated T cells, such as CD4+ and CD8+ T cells, impairs their ability to mount an effective immune response [54]. It binds to its ligands, including MHC class II, FGL1, and the TCR–CD3 complex, triggering the release of cytokines that inhibit T cell activation [55]. LAG3 expression has been found to be significantly elevated in HPV-associated tumors, particularly oropharyngeal cancers, alongside PD1 and CTLA4 [56]. Interaction with these immune checkpoints contributes to an immunosuppressive TME, which promotes tumor growth [57].

T cell immunoglobulin and mucin domain-3 (TIM3), a checkpoint protein expressed on various immune cells, also plays a critical role in immune evasion. The TIM3/Galectin-9 pathway is particularly relevant in the context of HPV-associated cancers, especially cervical cancer. Elevated levels of TIM3 and its ligand, Galectin-9, promote the accumulation of Tregs, which inhibit Th1 and CD8+ T cell activity, thereby facilitating tumor growth [58]. Galectin-9 expression has also been found to be increased in HPV-associated oropharyngeal cancers, contributing to the suppression of interferon gamma and cytokines such as IL-10 and IL-12. This suggests that high TIM3 expression may help tumors enhance Gal-9 expression on CD4+ T cells, making it a potential target for immunotherapies aimed at blocking these molecules to effectively treat patients with HPV-associated cancers [59].

Finally, the T cell immunoreceptor with immunoglobulin and immunoreceptor tyrosine-based inhibitory motif domain (TIGIT), expressed on CD8+ and CD4+ T cells, as well as natural killer (NK) cells, interacts with its ligand CD155 on tumor cells, suppressing immune function and promoting immune evasions. This mechanism has been observed in cervical cancer, where blocking the TIGIT-CD155 pathway restores CD8+ T cell function and inhibits tumor growth in vivo [60]. Elevated TIGIT expression has also been studied in oropharyngeal cancers, correlating with poor overall survival [61].

## 4. Current Monotherapy and Combination Therapies Involving Immune Checkpoint Inhibitors

### 4.1. Cervical Cancer

For patients with cervical cancer confined to the pelvis or without nodal disease, the recommended treatment is pelvic external beam radiation therapy (EBRT) with concurrent platinum-based chemotherapy and brachytherapy [62,63,64]. Preferred chemotherapy regimens include weekly cisplatin, carboplatin (if cisplatin-intolerant), or cisplatin/fluorouracil administered every 3 to 4 weeks during radiation therapy (RT). Pembrolizumab may be added for patients with FIGO 2014 stage III–IVA disease, based on data from the KeyNote-A18 trial (NCT04221945).

For patients with para-aortic lymph node involvement (FIGO 2018 stage IIIC2p), further imaging is recommended to rule out distant metastases. If no distant disease is found, treatment includes extended-field EBRT with concurrent platinum-based chemotherapy (cisplatin or carboplatin, if intolerant) and brachytherapy. Pembrolizumab may also be added for patients with FIGO stage III–IVA disease undergoing chemoradiation.

Cisplatin/paclitaxel or carboplatin/paclitaxel (with or without bevacizumab) is the preferred first-line treatment for PD-L1-positive (Combined Positive Score (CPS) ≥1) recurrent or metastatic cervical cancer, as supported by the Keynote-826 trial. The NCCN continues to recommend pembrolizumab in patients with PD-L1-positive tumors based on updated data confirming sustained survival benefits (median OS: 28.6 vs. 16.5 months). Bevacizumab combined with platinum-based chemotherapy (cisplatin/paclitaxel or topotecan/paclitaxel) was validated by the GOG 240 trial, which demonstrated improved OS (16.8 vs. 13.3 months) but with increased toxicities. Despite this, bevacizumab remains a category 1 option due to its survival benefits.

Pembrolizumab is FDA-approved for PD-L1-positive cervical tumors (CPS ≥ 1) that have progressed after chemotherapy. It is also used for MSI-H/dMMR or TMB-H solid tumors lacking other treatment options. Cemiplimab showed OS benefits (12 vs. 8.5 months) in recurrent/metastatic cervical cancer patients in the EMPOWER-Cervical-1 trial, making it a preferred second-line therapy. Single-agent chemotherapy options (e.g., paclitaxel, gemcitabine, and irinotecan) remain available, but response rates are generally low. Nivolumab is included as a “useful in certain circumstances” option for PD-L1-positive tumors, based on CheckMate-358 data.

In summary, the specific guidelines for adding ICIs to conventional treatments for cervical cancer include the use of pembrolizumab with CCRT for locally advanced disease and with chemotherapy, with or without bevacizumab, for recurrent or metastatic disease, as recommended by the NCCN and supported by FDA approval [23].

### 4.2. Anal Cancer

Phase II trial of nivolumab in refractory metastatic anal cancer showed a 24% response rate, while the KEYNOTE-028 trial of pembrolizumab in PD-L1-positive cases had a 17% response rate and a 58% disease control rate. The KEYNOTE-158 trial investigated pembrolizumab in non-colorectal microsatellite instability-high (MSI-H)/deficient mismatch repair (dMMR) cancers, including patients with anal cancers, showing an 11% response rate overall and a 15% rate in PD-L1-positive patients. Toxicities were manageable, with 13–17% experiencing grade 3 adverse events.

Based on these findings, the National Comprehensive Cancer Network (NCCN) guidelines recommend nivolumab and pembrolizumab for metastatic anal cancer after first-line chemotherapy, regardless of the MSI/MMR status, as PD-L1 expression and high tumor mutational load may contribute to their efficacy. This recommendation is grounded in clinical evidence demonstrating that these agents improve progression-free survival (PFS) and overall survival (OS) in this population [65].

### 4.3. Vaginal Cancer

Pembrolizumab + chemotherapy ± bevacizumab is a preferred first-line treatment for PD-L1-positive recurrent/metastatic vaginal cancer, extrapolated from cervical cancer guidelines. The KEYNOTE-826 trial (617 patients) showed that adding pembrolizumab to chemotherapy improved PFS (10.4 vs. 8.2 months; HR 0.65) and OS (50.4% vs. 40.4% at 24 months; HR 0.67). The response rate in PD-L1-positive tumors was 68.1% vs. 50.2% in the placebo group. Platinum-based chemotherapy (cisplatin/carboplatin + paclitaxel ± bevacizumab) is another preferred option based on the GOG 240 trial, which showed an OS benefit (16.8 vs. 13.3 months; *p* = 0.007) with bevacizumab.

Pembrolizumab is FDA-approved for PD-L1-positive, MSI-H/dMMR, or TMB-H tumors after chemotherapy, based on the KEYNOTE-028 and KEYNOTE-158 trials. Cemiplimab (PD-1 inhibitor) is a preferred second-line therapy based on the Empower-Cervical-1 trial, which showed an OS benefit (12 vs. 8.5 months; HR 0.69) over chemotherapy. Subgroup analysis indicated an OS benefit in SCC (11.1 vs. 8.8 months; HR 0.73) and adenocarcinoma (13.3 vs. 7 months; HR 0.56). Patient-reported outcomes favored cemiplimab over chemotherapy in terms of quality of life, pain, and appetite loss. Immunotherapy significantly improves survival and quality of life in recurrent/metastatic vaginal cancer.

The best available protocol for adding ICIs to conventional treatments for HPV-associated vaginal cancer involves the use of pembrolizumab for PD-L1-positive tumors, as recommended by the NCCN guidelines, and considering combination strategies with chemotherapy or radiotherapy based on extrapolated data from other HPV-associated cancers due to the rarity of vaginal cancer and the limited number of specific studies available [66].

### 4.4. Penile Cancer

Cisplatin-based chemotherapy (TIP: paclitaxel, ifosfamide, and cisplatin or 5-FU + cisplatin) is recommended as first-line therapy for penile cancer and may facilitate curative resection. The HERCULES (LACOG 0218) trial showed the potential efficacy of platinum-based chemotherapy + pembrolizumab followed by pembrolizumab maintenance, with an overall response rate (ORR) of 39.4%. Options include subsequent-line systemic therapy, RT for local control, or best supportive care. Checkpoint inhibitors (e.g., pembrolizumab) showed a median overall survival (OS) of 9.8 months in advanced or metastatic penile cancers and a higher response (35%) in cases of lymph node-only metastases. More clinical trials are strongly recommended due to limited second-line treatment data.

The optimal approach for incorporating ICIs into standard treatments for HPV-associated penile cancer involves using PD-1 inhibitors such as pembrolizumab, nivolumab, or cemiplimab, particularly in advanced or metastatic cases. For advanced penile squamous cell carcinoma (PSCC), the Global Society of Rare Genitourinary Tumors has noted that ICIs demonstrate activity in a subset of patients. Pembrolizumab monotherapy, a combination of nivolumab and ipilimumab, and cemiplimab monotherapy have been shown to be effective, with a median overall survival (OS) of 9.8 months and progression-free survival (PFS) of 3.2 months [67].

### 4.5. Vulvar Cancer

There is no established standard systemic therapy regimen for advanced or recurrent/metastatic vulvar cancer. However, several first-line treatment options are recommended by the NCCN Panel, including cisplatin/paclitaxel, carboplatin/paclitaxel, cisplatin/paclitaxel/bevacizumab (category 2B). Other recommended regimens include single-agent cisplatin and carboplatin.

Cemiplimab can be added as a second-line therapy option due to its efficacy observed in cervical cancer and advanced cutaneous SCC. A Phase II trial showed a response in 28 out of 59 patients with metastatic cutaneous SCC. In the Phase III Empower-Cervical-1 trial, cemiplimab significantly improved OS and PFS compared to chemotherapy (median OS of 12 months vs. 8.5 months; HR 0.69).

Pembrolizumab is a PD-1 inhibitor recommended for PD-L1-positive advanced or recurrent/metastatic vulvar cancer. In the Phase II KEYNOTE-158 trial, 10.9% of patients with advanced vulvar SCC responded to pembrolizumab, with a median OS of 6.2 months. Pembrolizumab showed an overall response rate (ORR) of 10.9% in vulvar SCC patients, with better responses in PD-L1-positive and TMB-H subgroups. In the KEYNOTE-158 trial, patients with TMB-H vulvar cancer had an ORR of approximately 17%, while non-TMB-H patients had a response rate of only 3.4%.

Another PD-1 inhibitor, nivolumab, showed some efficacy in a small cohort of HPV-positive or unknown-HPV-status vulvar cancer patients. The CheckMate 358 trial reported 12- and 18-month OS rates of 40% and 20%, respectively, and a 6-month PFS of 40%. Nivolumab is considered for HPV-related vulvar cancer in specific cases.

Given the limited data on subsequent therapies, the NCCN Panel strongly recommends participation in clinical trials or the use of pembrolizumab (for tumors with MSI-H/dMMR or TMB-H) for patients with advanced or recurrent/metastatic vulvar cancer.

In summary, the best available protocol for adding ICIs to conventional treatments for HPV-associated vulvar cancer includes using pembrolizumab as a second-line therapy for PD-L1-positive advanced or recurrent/metastatic disease, as recommended by the NCCN guidelines, and considering combination strategies with cisplatin and radiation therapy for locally advanced or unresectable cases [66]. All currently recommended monotherapies and combination ICI therapies for HPV-associated cancers, as per NCCN guidelines, are summarized in Table 2 for reference.

## 5. Efficacy of Immune Checkpoint Blockade Therapies in HPV-Associated Cancers

### 5.1. Cervical, Vaginal, and Vulvar Cancers

The KEYNOTE-028 trial in 2016 was the first to show modest efficacy with immune checkpoint inhibitors in metastatic or recurrent cervical cancer with pembrolizumab. A total of 24 patients were enrolled, with 20/24 showing PD-L1 positivity with a modified proportion score (MPS) ≥ 1%. According to Frenel et al. [3], the overall response rate (ORR) was 17% (95% CI, 5–37%), with a median duration of response (DOR) of 5.4 months (95% CI, 4.1–7.5 months) for the patients who achieved a partial response.

The CheckMate358 and EMPOWER-Cervical-1 trials investigated the effectiveness of nivolumab and cemiplimab, respectively, in the treatment of advanced cervical cancer with metastatic disease. The median PFS was 5.1 months (95% CI, 1.9–9.1), while the median OS reached 21.6 months (95% CI, 8.3–46.9) in this population.

In the EMPOWER trial, 608 patients were randomly assigned to treatment groups, irrespective of their PD-L1 status. Half of the patients received cemiplimab, while the remaining half were treated with one of the standard chemotherapy agents for cervical cancer. Patients receiving cemiplimab monotherapy showed significantly improved OS compared to those undergoing chemotherapy in both the SCC (10.9 months vs. 8.8 months; HR, 0.69; 95% CI, 0.56–0.85; *p* = 0.0023) and adenocarcinoma (13.5 months vs. 7.0 months; HR, 0.54; 95% CI, 0.36–0.81) subgroups [8].

The MK-3475-826 trial [73] was one of the first to show the advantage of adding pembrolizumab to chemotherapy. From a cohort of 617 subjects, one group received pembrolizumab and the other received a placebo every three weeks for 35 cycles with platinum-based chemotherapy for up to six cycles, with the option of bevacizumab. After a follow-up of 22.0 months, 548 patients with a PD-L1 CPS ≥ 1% showed a median PFS of 10.4 months in the pembrolizumab group, compared to 8.2 months in the placebo group (HR: 0.62; 95% CI: 0.50–0.77; *p* < 0.001). A follow-up analysis suggested that the use of bevacizumab (administered to 63% of patients) might have positively impacted the OS (HR: 0.63; 95% CI: 0.47–0.87).

### 5.2. Anal Cancer

Retifanlimab, a monoclonal antibody targeting PD-1, was evaluated in an open-label multicenter Phase II trial (POD1UM-202) involving patients with squamous cell carcinoma of the anal canal (SCCA) who had progressed following platinum-based chemotherapy. The study enrolled 94 patients, with a median follow-up period of 7.1 months [33] The treatment demonstrated an objective response rate (ORR) of 13.8%, including a complete response (CR) of 1.1% and a partial response (PR) of 12%, regardless of PD-1 expression. Additionally, 35.1% of patients achieved stable disease, resulting in a disease control rate (DCR) of 48.9%. The median PFS was 2.3 months, while the median overall survival (OS) reached 10.1 months.

The ECOG-ACRIN 2165 trial currently assesses disease-free survival (DFS) in patients with high-risk and localized anal cancer who have undergone definitive chemoradiation therapy, comparing outcomes between those receiving nivolumab and those under observation (NCT03233711). In parallel, the German Anal Cancer Group is leading the RADIANCE trial, which is designed to evaluate DFS in patients treated with concurrent chemoradiation plus durvalumab versus chemoradiation alone. The findings from both studies remain pending at the time of this review [74].

Another PD-L1 inhibitor, avelumab, has been investigated in patients with advanced or recurrent SCCA who had received prior treatment. The CARCAS trial (NCT03944252), a randomized Phase II, open-label, and multicenter study, assessed avelumab with or without cetuximab. In the avelumab-only group, the ORR was 10%, with three patients achieving a partial response (PR). The disease control rate (DCR) was 50%, while the median PFS and OS were 2.0 months and 13.9 months, respectively [31]. Overall, immune checkpoint inhibitor (ICI) monotherapy has shown a modest response rate of 10–14% in reported trials. However, in patients who responded, the responses were long-lasting, often extending beyond two years. Research efforts continue to focus on identifying biomarkers that predict long-term benefits from immunotherapy across different malignancies [75].

### 5.3. Penile Cancer

The HERCULES (LACOG 0218) trial was the first Phase II study to assess the combination of platinum-based chemotherapy and an immune checkpoint inhibitor in penile cancers. This single-arm trial, conducted across eleven centers in Brazil, involved 37 patients with locally advanced, recurrent, or metastatic cancer who received a regimen of platinum-based chemotherapy and pembrolizumab [36]. Among the 33 patients evaluated for efficacy, the objective response rate was 39.4%, which included 1 complete response and 12 partial responses. The response rate was higher in patients with tumors exhibiting high tumor mutational burden (TMB), reaching 75% (three of four patients). Additionally, HPV16-positive tumors showed a better response rate (55.6%) compared to those that were HPV-negative (35.0%). With a median follow-up of 24 months, the median PFS was 5.4 months (95% CI, 2.7–7.2), and the median OS was 9.6 months (95% CI, 6.4–13.1). The study also identified common genomic alterations in the cohort, including TP53 (57.1%), CDKN2A (51.4%), and TERT (31.4%).

## 6. Prognostic and Predictive Biomarkers for ICIs in HPV-Associated Cancers

Prognostic and predictive biomarkers for immune checkpoint inhibitors (ICIs) in HPV-associated cancers are critical for guiding treatment decisions and improving patient outcomes. HPV status serves as a key prognostic marker, with HPV-positive tumors generally exhibiting better clinical outcomes than HPV-negative ones [76,77,78]. Additionally, PD-1 and PD-L1 expression in tumor-infiltrating lymphocytes and tumor cells, respectively, are favorable prognostic markers, correlating with improved clinical outcomes in HPV-associated cancers [79].

Predictive biomarkers for ICI response include tumor mutational burden (TMB), which is often elevated in HPV16/18-positive pSCC cases and associated with enhanced responsiveness to ICIs [76]. PD-L1 expression, assessed using immunohistochemistry and the Combined Positive Score (CPS), is another key predictor of ICI efficacy, as recommended by the American Society of Clinical Oncology (ASCO) for HNSCC [80]. Moreover, HPV-positive tumors frequently exhibit an inflamed immune microenvironment enriched with CD3+, CD4+, CD8+, and PD-1+ cells, which has been linked to favorable ICI responses [81,82].

## 7. Challenges in the Use of Immune Checkpoint Blockers in HPV-Associated Lower Genital Tract Cancers

Immune checkpoint inhibitors (ICIs) have shown promise in treating HPV-associated lower genital tract cancers and anal cancer, but they face several challenges, including variable response rates, resistance, and adverse effects.

### 7.1. Variable Response Rates

The response rates to immune checkpoint inhibitors (ICIs) in HPV-associated cancers remain modest, with significant variability. A Phase II trial combining MEDI0457 and durvalumab in HPV-16/18-associated cancers reported an overall response rate (ORR) of 21%, while single-agent anti-PD1/PD-L1 therapies in anal squamous cell carcinoma (SCCA) have shown response rates of approximately 10–15% [83].

One major contributor is the heterogeneity in PD-L1 expression, which varies both spatially and temporally within tumors, complicating patient selection for ICIs [84]. PD-L1 expression is also influenced by regulatory mechanisms at genetic, transcriptional, and post-transcriptional levels, including oncogenic pathways such as JAK/STAT and PI3K/AKT [85,86]. Additionally, differences in immunohistochemical assays used to assess PD-L1 status, such as SP142, 28–8, 22C3, and SP263, further contribute to discrepancies in patient stratification [87,88].

Another key factor is HPV integration into the host genome, which decreases viral antigen expression and reduces immune recognition. This integration downregulates viral proteins like E6 and E7, impairing antigen presentation and immune response through multiple mechanisms. E5 inhibits the immunoproteasome, reducing MHC class I antigen presentation [89], while E7 upregulates SUV39H1, leading to epigenetic silencing of immune sensors like RIG-I and cGAS [90]. Additionally, E6 represses IFN-κ transcription, suppressing interferon-stimulated genes [91]. The subsequent loss of MHC class I expression further hampers the immune system’s ability to recognize and target HPV-infected cells, reducing the efficacy of ICIs and contributing to lower response rates in HPV-associated lower genital tract cancers and anal cancer [92].

### 7.2. Resistance Mechanisms

Resistance to immune checkpoint inhibitors (ICIs) represents a significant challenge in HPV-associated cancers, driven by multiple mechanisms. One key factor is the loss of Major Histocompatibility Complex (MHC) Class I expression, which is essential for presenting antigens to CD8+ T cells. The downregulation or complete loss of MHC Class I impairs antigen presentation and promotes immune evasion, a phenomenon observed in HPV-associated cancers such as cervical and vulvar neoplasia, where a substantial proportion of tumors exhibit clonal or complete loss of MHC Class I expression [92].

The HPV E5 oncoprotein further exacerbates this resistance by downregulating MHC Class I expression and interfering with antigen presentation. E5 inhibits the immunoproteasome and the STING pathway, both of which are critical for effective antigen processing and presentation, thereby suppressing immune responses and contributing to ICI resistance.

Additionally, tumors can evade immune surveillance by upregulating alternative immune checkpoints (AICs) such as lymphocyte-activation gene 3 (LAG-3), T cell immunoglobulin and mucin-domain containing-3 (TIM-3), and T cell immunoreceptor with Ig and ITIM domains (TIGIT), which inhibit T cell function and contribute to immune escape, thereby reducing the efficacy of immune checkpoint inhibitors (ICIs) [52]. AICs extend beyond the conventional PD-1, PD-L1, and CTLA-4 checkpoints, and their upregulation is often observed in patients with poor responses to ICIs. These include members of the B7 family, such as B7-H3 (CD276), B7-H4 (B7x), and V-domain immunoglobulin suppressor of T cell activation (VISTA), as well as molecules like indoleamine 2,3-dioxygenase 1 (IDO1) [93,94]. In HPV-associated cancers, tumors can adapt to primary checkpoint blockade by upregulating alternative pathways, with IDO1 frequently elevated in HPV-positive tumors, leading to T cell suppression and immune tolerance [94]. Furthermore, TIM-3 upregulation has been observed in T cells resistant to PD-1 blockade, indicating its role as a compensatory inhibitory pathway [95]. These AICs contribute to immune evasion by inhibiting T cell activation and proliferation, reducing cytokine production, and promoting T cell exhaustion, highlighting the need for combination therapies targeting multiple checkpoints to overcome resistance and enhance immunotherapy efficacy in HPV-associated cancers.

The immunosuppressive tumor microenvironment (TME) further compounds resistance as HPV-associated cancers often harbor regulatory T cells (Tregs), myeloid-derived suppressor cells (MDSCs), and immunosuppressive cytokines such as Transforming Growth Factor-beta (TGF-β) and Interleukin-10 (IL-10). These components create an immunosuppressive milieu that suppresses effective anti-tumor immune responses, with MDSCs, in particular, inhibiting the activation and function of CD8+ T cells, thereby reinforcing resistance to ICIs [96].

### 7.3. Adverse Effects

Adverse events associated with immune checkpoint inhibitors (ICIs) are a critical consideration in HPV-associated cancers, with severe toxicities necessitating careful monitoring and management. In the Phase II trial of MEDI0457 and durvalumab, 23% of participants experienced grade 3–4 treatment-related adverse events, underscoring the potential severity of immune-related toxicities [83].

Common immune-related adverse events (irAEs) include colitis, dermatitis, pneumonitis, and endocrinopathies such as thyroiditis and adrenal insufficiency, many of which require immunosuppressive therapy or even treatment discontinuation [97]. For example, immune-mediated colitis can present with severe diarrhea and often requires high-dose corticosteroids or other immunosuppressants if refractory to initial treatment [98]. Pneumonitis, another serious irAE, carries a risk of fatality and typically necessitates systemic corticosteroids for management [99,100]. Additionally, chronic irAEs can significantly impact a patient’s quality of life, with conditions such as inflammatory arthropathy or peripheral neuropathy leading to persistent pain and discomfort [101]. Endocrinopathies like adrenal insufficiency may result in long-term complications, including osteoporosis, weight gain, and impaired wound healing, necessitating lifelong management [102].

The American Society of Clinical Oncology (ASCO) guidelines recommend routine follow-ups to assess for late-onset irAEs, with evaluations of clinical chemistries, liver enzymes, and thyroid function, as well as collaboration with subspecialists, to address the broad spectrum of irAE-related complications [97]. The findings from the MEDI0457 [103] highlight the substantial burden of severe toxicities, reinforcing the need for a comprehensive approach that includes prompt immunosuppressive intervention, diligent long-term surveillance, and multidisciplinary care to optimize patient outcomes.

### 7.4. Challenges in Combination Strategies

Combination strategies aimed at enhancing the efficacy of immune checkpoint inhibitors (ICIs) in HPV-associated cancers face several significant challenges. The combination of ICIs with chemotherapy has shown improved outcomes, as demonstrated in the KEYNOTE-826 trial evaluating pembrolizumab with chemotherapy in cervical cancer. However, toxicity remains a major concern as the addition of chemotherapy can exacerbate immune-related adverse events (irAEs) and increase the overall treatment burden, often necessitating dose adjustments or discontinuation of therapy [104,105,106].

Similarly, combining ICIs with radiotherapy is an area of interest, particularly due to the potential for the “abscopal effect”, in which localized radiotherapy induces systemic anti-tumor immune responses. However, this effect remains unpredictable and inconsistent, limiting its clinical applicability. Moreover, the combination of radiotherapy with ICIs can heighten the risk of irAEs, such as radiation pneumonitis, further complicating treatment strategies [107,108,109,110].

Another promising approach involves combining ICIs with therapeutic HPV vaccines to leverage synergistic immune responses. However, determining the optimal sequencing and dosing remains a challenge as improper timing could lead to suboptimal immune activation or increased toxicity. While ongoing studies are investigating these combinations, additional data are required to establish standardized protocols that balance efficacy and safety [111,112,113].

In summary, although combination strategies hold significant promise in improving ICI efficacy in HPV-associated cancers, they also present substantial challenges related to toxicity, variability in response, and the need for optimized treatment protocols.

### 7.5. Cost and Accessibility

The cost and accessibility of immune checkpoint inhibitors (ICIs) for HPV-related cancers pose significant challenges, particularly in low- and middle-income countries (LMICs). ICIs such as pembrolizumab and nivolumab are prohibitively expensive, making them largely inaccessible in these regions [114,115]. The high cost of these therapies, coupled with limited healthcare infrastructure, significantly restricts their widespread use. A review in the journal Cancer highlights that the excessive cost and lack of affordability of new cancer treatments are among the most substantial barriers in low-income countries [115]. Additionally, logistical constraints, including the lack of specialized healthcare facilities and trained personnel, further hinder access to these advanced therapies [116,117,118].

Another major issue is disparities in clinical trial enrollment, which disproportionately affect marginalized populations, including racial and ethnic minorities and those from rural areas. The American Society of Clinical Oncology (ASCO) has reported that these groups are underrepresented in clinical trials, impacting the generalizability of trial results and the development of equitable treatment guidelines [119,120]. For instance, Black and Hispanic patients remain significantly underrepresented in clinical trials for gynecologic malignancies, including HPV-related cervical cancer [121,122].

Addressing these disparities requires increased community engagement, improved access to trial information, and modifications in trial design to enhance inclusivity. Organizations such as the American Association for Cancer Research, the American Cancer Society, the ASCO, and the National Cancer Institute advocate for greater involvement of minority healthcare teams in study design and the removal of cost barriers to trial participation [123,124,125].

### 7.6. Regulatory and Clinical Trial Challenges

The regulatory and clinical trial challenges associated with immune checkpoint inhibitors (ICIs) for HPV-related cancers are multifaceted, limiting their widespread clinical adoption. One major hurdle is the lack of large-scale Phase III trials as most available data come from Phase I and II studies. While these early trials show promise, they do not provide the robust evidence required for regulatory approval and widespread clinical implementation. This gap is particularly critical for cancers like cervical cancer, where ICIs such as pembrolizumab have demonstrated potential but require further validation in larger definitive studies [126].

Another challenge is the need for immune-related response criteria (iRECIST) as traditional Response Evaluation Criteria in Solid Tumors (RECIST) often fail to account for unique ICI-associated response patterns, such as pseudoprogression and hyperprogression [127,128,129]. iRECIST allows continued treatment of patients who initially appear to progress but may later show a response, providing a more accurate assessment of ICI efficacy. The American Society of Clinical Oncology and the Society for Immunotherapy of Cancer recommend implementing iRECIST in clinical trials to enhance immunotherapy evaluation [130].

Regulatory challenges further complicate the approval process as regulatory bodies require strong evidence from well-designed clinical trials. Given the unique response kinetics of ICIs, novel trial designs and endpoints—such as milestone analysis and restricted mean survival time (RMST)—are necessary to adequately capture their long-term benefits. Additionally, integrating predictive biomarkers like PD-L1 expression is crucial for patient selection and improving trial reliability [131,132].

## 8. Conclusions

The efficacy of ICIs in HPV-associated cancers highlights the critical need to understand the tumor microenvironment and the ways in which HPV causes immune evasion. Ongoing research and clinical trials are essential to further elucidate these mechanisms and optimize immunotherapeutic strategies for patients with HPV-related malignancies. The integration of ICIs into standard treatment protocols represents a significant advancement in the management of these cancers, offering new hope for better patient outcomes.

## Figures and Tables

**Table 1 cancers-17-01260-t001:** Clinical trials for HPV-associated cancer treatments.

Study	Clinical Trial Number	Type of Cancer	Condition	Phase	Intervention	Sample Size	Outcome
KEYNOTE-158 [2]	NCT02628067	Cervical Cancer	Advanced Cervical Cancer	II	Pembrolizumab 200 mg/3 weeks	98	ORR = 12.2% (95% CI:6.5–20.4)
Keynote-028 [3]	NCT02054806	Cervical Cancer	Advanced PD-L1-positive cervical cancer	Ib	Pembrolizumab 10 mg/kg/2 weeks	24	ORR = 17% (95% CI:5–37)
NRG-GY002 [4]	NCT02257528	Cervical Cancer	Persistent, recurrent, or metastatic cervical cancer	II	Nivolumab 3 mg/kg/2 weeks	26	ORR = 4% (90% CI: 0.4–22.9)
[5]	-	Cervical Cancer	Advanced or recurrent cervical cancer	II	Nivolumab 240 mg/2 weeks	20	ORR = 25% (80% CI: 13–41)
Checkmate-358 [6]	NCT02488759	Cervical Cancer	Recurrent or metastatic cervical, vaginal, or vulvar carcinoma	I/II	Nivolumab 240 mg/2 weeks	19	ORR = 26.3% (95% CI: 9.1–51.2)
[7]	-	Cervical Cancer	Recurrent or metastatic cervical cancer	I	Cemiplimab 3 mg/kg/2 weeks or cemiplimab +hFRT	10	ORR = 10% (95% CI: 0.3–44.5)
EMPOWER [8]	NCT03257267	Cervical Cancer	Recurrent cervical cancer	III	Cemiplimab 350 mg/3 weeks or chemotherapy	608	ORR = 16.4% (95% CI, 12.5–21.1)
[9]	-	Cervical Cancer	Recurrent or metastatic cervical cancer	II	Balstilimab 3 mg/kg/week	140	ORR = 15% (95% CI: 10.0–21.8)
IND221 [10]	-	Cervical Cancer	Advanced, recurrent, or metastatic gynecologic cancers	I	Monalizumab 10 mg/kg/2 weeks	9	ORR = 0%
[11]	NCT02517398 NCT03427411	Cervical Cancer	Human papillomavirus-associated malignancies	I/II	Bintrafusp alfa/2 weeks	10	ORR = 30.5% (95% CI, 19.2–43.9
[12]	NCT04246489	Cervical Cancer	Platinum-experienced cervical cancer	II	Bintrafusp alfa 1200 mg/IV/2 weeks	146	ORR = 21.9% (95% CI, 15.5–29.5)
[13]	NCT03444376	Cervical Cancer	Advanced cervical cancer	II	2 mg GX-188E/IM + pembrolizumab 200 mg/3 weeks	36	ORR = 42% (95% CI: 23–63) PD-L1 (+) group: ORR = 50% (95% CI:27–73)
[14]	NCT02471846	Cervical Cancer	Advanced solid tumors	II	Navoximod/12 h + atezolizumab 1200 mg/IV/3 weeks	157	Dose Escalation (n = 66) ORR = 9%; Dose Expansion (n = 92): ORR = 11%
Checkmate-358 [15]	NCT02488759	Cervical Cancer	Recurrent or metastatic cervical cancer	I/II	Nivolumab 3 mg/kg/2 weeks + ipilimumab 1 mg/kg/6 weeks OR nivolumab 1 mg/kg/3 weeks + ipilimumab 3 mg/kg/3 weeks	176	ORR = 26% (95% CI: 9–51) with nivolumab, 31% (95% CI: 18–47) with NIVO3+IPI1, 40% (95% CI: 26–56) with randomized NIVO1+IPI3, and 38% (95% CI: 29–48) with pooled NIVO1+IPI3
C550 [16]	NCT03495882	Cervical Cancer	Advanced cervical cancer	II	Balstilimab 3 mg/kg/2 weeks + zalifrelimab 1 mg/kg/6 weeks	155	ORR = 25.6% (95% CI: 18.8 to 33.9)
[17]	-	Cervical Cancer	Persistent, recurrent, or metastatic cervical cancer	I	TILs + nivolumab 3 mg/kg	80	ORR= 25%
AdvanTIG-202 [18]	NCT04693234	Cervical Cancer	Recurrent or metastatic cervical cancer	II	TIS 200mg/IV/3 weeks + OCI 900mg/IV/3 weeks OR TIS 200mg/IV/3 weeks	178	Cohort1: PD-L1(-): ORR = 22.5% (95% CI: 15.8–30.3) PD-L1(+): ORR = 26.2% (95% CI: 17.2–36.9) Cohort2: ORR = 32.5%
COMPASSION-13 [19]	NCT04868708	Cervical Cancer	Persistent, recurrent, or metastatic cervical cancer	II	Cohort A-15: (cadonilimab 15 mg/kg/3 weeks + chemotherapy), cohort A-10: (cadonilimb 10 mg/kg/3 weeks + chemotherapy), Cohort B-10: (cadonilimab 10 mg/kg/3 weeks + chemotherapy and bevacizumab)	45	A-15: ORR = 73.3%; A-10: ORR = 68.8%; B-10: ORR = 92.3%
[20]	NCT02921269	Cervical Cancer	Advanced cervical cancer	II	Bevacizumab 15 mg/kg/IV/3 weeks + atezolizumab 1200 mg/IV/3 weeks	10	ORR = 0%
[21]	NCT04221945	Cervical Cancer	Newly diagnosed, high-risk, and locally advanced cervical cancer	III	Chemotherapy + pembrolizumab 200 mg/3 weeks	1060	36-month OS = 82.6% in the pembrolizumab–chemoradiotherapy group and 74.8% in the placebo–chemoradiotherapy group
CALLA study [22]	NCT03830866	Cervical Cancer	Locally advanced cervical cancer	III	Chemotherapy + durvalumab 1500 mg/IV/4 weeks	770	12-month PFS = 76% (71·3–80·0) with durvalumab and 73·3% (68·4–77·5) with placebo
Keynote-826 [23]	NCT03635567	Cervical Cancer	Persistent, recurrent, or metastatic cervical cancer	III	Chemotherapy + pembrolizumab (200 mg) or placebo every 3 weeks	548	PFS = pembrolizumab group: 10.4; placebo group: 8.2
PRIMMO study [24]	NCT03192059	Cervical Cancer	Persistent, recurrent, or metastatic cervical or endometrial carcinoma	II	Pembrolizumab/3 weeks + radiation + immunomodulatory cocktail (Vitamin D, aspirin, cyclophosphamide, and lansoprazole)	18	ORR = 11.1% (90% CI: 2.0–31.0)
[7]	-	Cervical Cancer	Recurrent or metastatic cervical cancer	I	Cemiplimab 3 mg/kg/IV/2 weeks OR cemiplimab 3 mg/kg/IV/2 weeks + hfRT	10	ORR = 10%
NICOL [25]	NCT03298893	Cervical Cancer	Locally advanced cervical cancer	I	Nivolumab + chemoradiotherapy	16	One-year PFS = 81.2% [95% CI: 64.2–100].
GOTIC-018 [26]	-	Cervical Cancer	Locally advanced cervical cancer	I	Nivolumab 240 mg/2 weeks + chemoradiotherapy	30	Cohort A: CR = 73.3%; PR = 26.7%; Cohort B: CR = 66.7%; PR = 33.3%
CLAP [27]	NCT03816553	Cervical Cancer	Advanced cervical cancer	II	Camrelizumab 200 mg/2 weeks + apatinib 250 mg	45	ORR = 55.6% (95% CI: 40.0–70.4); PD-L1(+) = 69%
KEYNOTE-158 [28]	NCT02628067	Vulvar Cancer	Advanced (metastatic and unresectable) vulvar SCC	II	Pembrolizumab 200 mg/3 weeks	101	ORR = 11% (95% CI: 6–19)
Keynote-028 [29]	NCT02054806	Vulvar Cancer	Advanced (unresectable and metastatic) vulvar SCC	Ib	Pembrolizumab 10 mg/kg/2 weeks	18	ORR = 6% (95% CI: 0–27)
Checkmate-358 [6]	NCT02488759	Vulvar/Vaginal Cancer	Recurrent or metastatic cervical, vaginal, or vulvar carcinoma	I/II	Nivolumab 240 mg/2 weeks	5	ORR = 20% (95% CI: 0.5–71.6)
NCI9673 [30]	NCT02314169	Anal Cancer	Treatment-refractory metastatic SCCA	II	Nivolumab every 2 weeks (3 mg/kg)	37	ORR = 24% (95% CI: 15–33)
CARACAS (arm 1) [31]	NCT03944252	Anal Cancer	Anal squamous cell carcinoma	II	Avelumab alone	30	ORR = 10% (95% CI: 2.5–26.5)
CARACAS (arm 2) [31]	NCT03944252	Anal Cancer	Anal squamous cell carcinoma	II	Avelumab combined with cetuximab	30	ORR = 17% (95% CI: 5.6–34.7)
KEYNOTE-028 [29]	NCT02054806	Anal Cancer	Anal squamous cell carcinoma	Ib	Pembrolizumab 10 mg/kg/2 weeks	25	ORR = 17% (95% CI: 5–37)
KEYNOTE-158 [32]	NCT02628067	Anal Cancer	Previously treated advanced anal squamous cell carcinoma	II	Pembrolizumab 200 mg/3 weeks	112	ORR = 11% (95% CI: 6–18)
POD1UM-202 [33]	NCT03597295	Anal Cancer	Anal squamous cell carcinoma	II	Retifanlimab 500 mg IV/4 weeks	94	ORR = 13.8% (95% CI: 7.6–22.5)
[34]	NCT03074513	Anal Cancer	Unresectable/metastatic anal cancer	II	Atezolizumab (1200 mg) and bevacizumab (15 mg/kg) IV/3 weeks	19	ORR = 11%
EPIC [35]	NCT95561634	Penile Cancer	Advanced penile cancer	II	Cemiplimab 350mg IV D1/3 weeks	18	ORR = 16.6% (95% CI: 5.8–39.2)
HERCULES (LACOG 0218) [36]	NCT04224740	Penile Cancer	Advanced penile cancer	II	5-FU 1000mg/m^2^/day IV + cisplatin 70mg/m^2^ (or carboplatin AUC 5) IV + pembrolizumab 200 mg IV/3 weeks	33	ORR = 39.4% (95% CI: 22.9–57.9)
[37]	NCT03333616	Penile Cancer	Rare genitourinary cancers, including penile cancer	II	Nivolumab 3 mg/kg and ipilimumab 1 mg/kg IV/3 weeks	5	ORR = 6% (80% CI: 1–20)
PERICLES [38]	NCT03686332	Penile Cancer	Stage IVa penile cancer	II	Atezolizumab 1200 mg/3 weeks + RT (arm 1)/atezolizumab 1200 mg/3 weeks—RT (arm 2)	30	ORR = 16.7% (95% CI: 6–35)
[39]	NCT02496208	Penile Cancer	Advanced/metastatic genitourinary tumors	I	Cabozantinib and nivolumab (CaboNivo) alone or with ipilimumab (CaboNivoIpi)	9	ORR = 4% (95% CI: 13.7–78.8)

**Table 2 cancers-17-01260-t002:** NCCN-recommended monotherapies and combination therapies for HPV-associated cancers using immune checkpoint inhibitors.

Cervical Cancer [68]	Anal Cancer [69]	Vaginal Cancer [70]
Locally advanced cancer (chemoradiation)		
Cisplatin + pembrolizumab: FIGO 2018 stage III–IVA		
Carboplatin + pembrolizumab in cisplatin-intolerant category 1: FIGO 2018 stage III–IVA		
Recurrent/metastatic cancer		
First-line therapy		
PD-L1-positive tumors: Pembrolizumab + cisplatin/paclitaxel ± bevacizumab or Pembrolizumab + carboplatin/paclitaxel ± bevacizumab	Carboplatin + paclitaxel + retifanlimab-dlwr	PD-L1-positive tumors: Pembrolizumab + cisplatin/paclitaxel ± bevacizumab or Pembrolizumab + carboplatin/paclitaxel ± bevacizumab
Second-line and subsequent therapy		
Pembrolizumab for TMB-H tumor or PD-L1-positive for MSI-H/dMMR tumors	Cemiplimab Dostarlimab Nivolumab Pembrolizumab Retifanlimab, tislelizumab, or toripalimab	Pembrolizumab for TMB-H, PD-L1-positive, or MSI-H/dMMR tumors
Other recommendations		Other recommended regimens
Cemiplimab		Cemiplimab
Useful in certain circumstances		Useful in certain circumstances
PD-L1-positive tumors—Nivolumab and tisotumab vedotin + pembrolizumab		PD-L1-positive tumors—Nivolumab
Penile cancer [71]	Vulvar cancer [72]	
First-line systemic therapy for metastatic/recurrent disease (other recommended regimens)	First-line therapy for advanced or recurrent/metastatic disease	
5-FU + cisplatin + pembrolizumab followed by pembrolizumab maintenance therapy	Pembrolizumab + cisplatin/paclitaxel ± bevacizumab	
5-FU + carboplatin + pembrolizumab followed by pembrolizumab maintenance therapy	Pembrolizumab + carboplatin/paclitaxel ± bevacizumab	
Subsequent-line systemic therapy for metastatic/recurrent disease	Second-line or subsequent therapy	
Clinical trial	Cemiplimab	
Pembrolizumab is indicated for patients with unresectable or metastatic tumors, including those with microsatellite instability-high (MSI-H) or mismatch repair-deficient (dMMR) tumors that have progressed after prior treatment, and for those with high tumor mutational burden (TMB-H, TMB ≥10 mut/Mb) who have exhausted previous treatment options.		
Useful in certain circumstances	Useful in certain circumstances (biomarker-directed therapy)	
Cetuximab	TMB-high (TMB-H)—Pembrolizumab PD-L1-positive—Pembrolizumab MSI-high (MSI-H)/MMR-deficient (dMMR) tumors—Pembrolizumab HPV-related tumor—Nivolumab	

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
