# Peer review of "Advances and Challenges in the Treatment of HPV-Associated Lower Genital Tract Cancers by Immune Checkpoint Blockers: Insights from Basic and Clinical Science"

_cancers, 2025, doi:10.3390/cancers17081260_

Round 1
Reviewer 1 Report
Comments and Suggestions for Authors

Author Response
Dear Reviewer,
Thank you for your valuable feedback on our manuscript titled “Advances and Challenges in the Treatment of HPV-Associated Lower Genital Tract Cancers by Immune Checkpoint Blockers: Insights from Basic and Clinical Science.”
We have carefully reviewed your comments and made the following revisions:
-
Updated Data on Incidence (Paragraph 2, Lines 47-66): We have incorporated more recent data on the incidence of HPV-associated cancers, as suggested, to ensure that the information is current and reflective of the latest findings.
-
Font Consistency: We have ensured that the font is uniform throughout the manuscript, as per your suggestion.
-
Simple Summary: We have added a Simple Summary section to the manuscript, in accordance with the manuscript requirements.
-
References: We have re-organized the references to be numbered in the order of their appearance in the text, including in the table captions and figure legends. Additionally, we have listed the references individually at the end of the manuscript, as requested.
We appreciate your careful review of our manuscript and the insightful comments you provided. We believe these revisions have strengthened the manuscript, and we are grateful for your time and effort in reviewing our work.
Thank you again for your valuable feedback. We look forward to your continued guidance and hope the revised manuscript meets your expectations.
Sincerely,
Marhama Zafar
Reviewer 2 Report
Comments and Suggestions for Authors
the research topic is very interesting and could bring benefit to the scientific community. the authors choose comprehensive approach to describe Human Papilloma Virus induced cancers according to the already published data. the article is well organized, easy to read and understand. the introduction briefly describes importance of the study and key points necessary for proper understanding of the literature review.
for better organization and understanding of the research the authors divided the article in few paragraphs. the second paragraph describe implication of the HPV on human health's and which cancers could originated form it. Also same paragraph underline the importance of the vaccination in reducing risk of the cancer developments.
the third paragraph analyze mechanism of cancer development and potential influence on it.
the fourth and fifth paragraph give information about available therapies and their impact on the quality of life for the cancers patient.
the conclusion summarize the described article and pointed out the most important conclusions
i have suggestion to authors make paragraph with description of the research topic!
what was the key word in the scientific search?
Which scientific database were analyzed?
add PRISM diagram about article selection.
Author Response
Dear Reviewer,
Thank you for your insightful feedback. We appreciate your suggestion to include a more detailed description of our research topic, along with the inclusion of methodological clarity regarding the selection of clinical trials for the summary table.
While this manuscript is intended as a narrative review rather than a systematic or scoping review, we agree that providing a brief explanation of our literature search strategy would improve transparency. To address this, we have added a paragraph to the Introduction outlining our search methodology, including the keywords used and the scientific databases analyzed.
Regarding the PRISMA diagram, since this is not a systematic review, we did not include a formal PRISMA flow diagram. However, we believe the additional clarification in the Introduction will sufficiently address your concerns and provide the necessary context for the reader.
We appreciate your thoughtful suggestions, which have undoubtedly helped strengthen the manuscript. Thank you again for your time and valuable input.
Sincerely,
Marhama Zafar
Reviewer 3 Report
Comments and Suggestions for Authors
Very informative and well-organized review article.
HPV-related cancers are common and represent a challenge.
ICI are very expensive and are not available in low resource settings.
This review would help clinicians and basic science researchers to understand the mechanism of action and the possible new directions in research and treatment.
Would suggest adding the best available protocol of adding ICI to conventional treatments.
Author Response
Dear Reviewer,
Thank you for your valuable suggestion regarding the inclusion of the best available protocol for adding immune checkpoint inhibitors (ICIs) to conventional treatments. We appreciate your insight, and we have incorporated the recommended changes into the manuscript.
We believe these revisions enhance the clarity and comprehensiveness of the content, and we appreciate your constructive feedback.
Sincerely,
Marhama Zafar
Round 2
Reviewer 2 Report
Comments and Suggestions for Authors
the authors made effort to responds to reviewers' suggestion in the appropriate manner and they done it adequately. the implemented changes improved article quality and now it fulfill minimum requirements for publication. presented analysis of literature could bring benefit to the scientific community especially regarding the HPV and prevention of disease originated from it..